# Controllable Shrinking Fabrication of Solid-State Nanopores

**DOI:** 10.3390/mi13060923

**Published:** 2022-06-10

**Authors:** Xin Lei, Jiayan Zhang, Hao Hong, Zhishan Yuan, Zewen Liu

**Affiliations:** 1School of Chemistry, Beihang University, Beijing 100191, China; leixin@buaa.edu.cn (X.L.); ZhangJiaYan@buaa.edu.cn (J.Z.); 2School of Integrated Circuits, Tsinghua University, Beijing 100084, China; honghao@tsinghua.edu.cn; 3Department of Microelectronics, Delft University of Technology, 2628 CD Delft, The Netherlands; 4School of Electro-Mechanical Engineering, Guangdong University of Technology, Guangzhou 510006, China

**Keywords:** solid-state nanopores, shrinking fabrication, size and shape control, high energy beam

## Abstract

Nanopores have attracted widespread attention in DNA sequencing and protein or biomarker detection, owning to the single-molecule-scale detection accuracy. Despite the most use of naturally biological nanopores before, solid-state nanopores are widely developed with strong robustness, controllable sizes and geometries, a wide range of materials available, as well as flexible manufacturing. Therefore, various techniques typically based on focused ion beam or electron beam have been explored to drill nanopores directly on free-standing nanofilms. To further reduce and sculpt the pore size and shape for nano or sub-nano space-time sensing precision, various controllable shrinking technologies have been employed. Correspondingly, high-energy-beam-induced contraction with direct visual feedback represents the most widely used. The ability to change the pore diameter was attributed to surface tension induced original material migration into the nanopore center or new material deposition on the nanopore surface. This paper reviews typical solid-state nanopore shrinkage technologies, based on the careful summary of their principles and characteristics in particularly size and morphology changes. Furthermore, the advantages and disadvantages of different methods have also been compared completely. Finally, this review concludes with an optimistic outlook on the future of solid-state nanopores.

## 1. Introduction

Since nanopores technology were first successfully used as sensors for the detection of individual polynucleotide molecules [1], nanopores had been brought more attractive prospects in various field, such as DNA sequencing [2], protein detection [3,4], drug screening [5], ion logic circuit [6] and energy conversion [7]. Currently, an abundant of review on nanopore [8], nanochannel [9] and ion channel [10] were presented from fabrication [11,12] to application [13,14]. Among them, the representative review and research from Dekker’s group pay more attention to solid-state nanopore preparation and molecular sensors [15]. For ion channel, Jiang’s group use chemically modified polyethylene terephthalate (PET) channel to establish asymmetric biomimetic ion transport system [16] and extended to ions and molecular separation [17], controllable gating [18], and clean energy conversion [19]. Recently, the replacement of silicon nitride by 2D materials represented by graphene is the hotspot of solid-state nanopores in energy conversion [20]. Furthermore, quantum-confined superfluid (QSF) [21,22] provide more opportunities in ion channel and nanopore for ions and molecules superfluid in a confined space without kinetic energy loss.

However, the size and shape of nanopore will directly affect the detecting accuracy of solid-state nanopore sensor in those applications. Therefore, controllable fabrication of solid-state nanopore still play a vital role. At first, researchers had sought to fabricate the solid-state nanopore directly in the free-standing nanomembrane by FIB drilling [23]. However, limited by the resolution of the gallium ion beam, beam diameter, beam shape and redeposition, the typical feature size of the hole is always above 10 nm [24]. Furthermore, chemical ways such as ion track etching [25,26], mask etching [27] and chemical solution etching [28,29] were explored to fabricate the solid-state nanopore directly in polymer membrane. Although the smallest diameter was down 2 nm [30], rough hole wall and the specific material requirement limit the widespread use. Recently, a simple fabrication method, called “dielectric breakdown” has gained attention [31]. The nanopore with 1.3 ± 0.6 nm (*n* = 23) was generated by the high electric field breakdown between two sides of the nanofilm [32]. However, the uncontrollable location of nanopore generation and the indistinguishable pore size increase and pore number increase still need to be solved.

Nevertheless, control over the pore size and shape was still limited with nanopore manufacturing efficiency. In order to achieve the controllable manufacturing, various shrinking techniques, typically based on focused ion beam or electron beam with visual feedback control, emerged to further reduce the pore size and modified the pore shape in nanoscale. The first shrinking strategy was fully designed and experimented by the ion-beam sculpting system at Harvard University in 2001 [23]. Initially the large-scale nanopore was drilled by high-flux focused ion beam (FIB), and then edge material migration at low-flux enabled nanometer control of structural dimensions from 61 nm to only 1.8 nm. However, the complicated adjustment of ion drilling and reducing process, as well as too fast dimensional change or even nanofilm destruction directly, limited it to a few FIB-equipped laboratories. To further improve the dimensional control accuracy, the most reliable approach of controllable manufacturing was presented by Dekker’s group in a transmission electron microscope (TEM) [33]. Under the radiation of high-energy electron beam (EB), nanopores could be opened and sculpted by edge material migration with single-nanometer precision. Real-time visual feedback in TEM also ensured the remarkable controllability and size adjustment. Based on that, Al_2_O_3_ nanopore with a diameter as small as 1 nm was achieved in sub-nanometer accuracy [34]. Furthermore, laser [35] or direct high-temperature heating [36] had also been proved as alternative thermal sources for the material migration during the controllable shrinking process.

Further development in high-energy-beam induced shrinkage were exploited to decompose and deposit different material on nanopore surface. In this way, pore size could be precisely adjusted in nanoscale by controlling the deposition time and the supply of raw materials. Among them, electron beam induced deposition (EBID) had been widely used owning to the accessible deposition material such as the hydrocarbon in chamber, and the higher dimensional control accuracy with visual feedback. Recently, EBID was successfully realized in scanning electron microscope (SEM) to fine turned the Si_3_N_4_ nanopore diameter from 90 nm to only 5.3 nm [37]. In addition, common deposition skills such as atomic layer deposition (ALD) [38] and chemical vapor deposition (CVD) [39] could also provide complete steps to fine-tune both the sizes and the surface properties of already-made nanopores. Hence, Nanopores in different materials have been successfully shrunk, not just silicon oxide [33] and silicon nitride [40] but crystalline materials such as magnesium and its alloys [41], particularly from the typical two-dimensional material graphene [42] to low-cost polymer [43].

Based on the above advantages, shrinking fabrication plays a key role in pore-size control and pore-shape modification. So, we review the most recent advances and older well-known works in nanopore shrinkage, which is divided into material migration and deposition as shown in Figure 1. Related characteristics in this field: how they work, changing in size and morphology, size-control accuracy, etc. have been briefly discussed and compared. Finally, we conclude with a positive outlook on solid-state nanopores.

## 2. Shrinkage of Nanopore Edge Material Migration

As always, the possibility of nanopore modification with sub-nanometer accuracy and sublime consistency is attractive to researchers. Fine-tuning of nanopore sizes as smaller as the feature diameter of the target biomolecule, usually no more than 5 nm, is vital in single-molecule detection or sequencing. The most use of shrinking strategies was attributed to transient fluidization and migration from the nanopore edge to the center under radiating of high-energetic beams such as FIB, EB, laser, or direct-thermal heating. In the following text, they have been discussed belonging in different classifications. The central performance is shown in Table 1.

### 2.1. EB-Induced Material Migration

The physics of witnessed nanopore shrinkage and expansion by high-energy electron beam radiation satisfies the principle of minimum surface free energy [33]. Exposing to consistent radiation, the material surrounding nanopores will transfer into a viscous state, after which the fluidized-material migration will occur along the nanopore edge determined by the surface tension with a lower surface free energy. The change in free energy ΔF can be expressed as:(1)$$\Delta F=\gamma \Delta A=2\pi \gamma \left(rh-r^{2}\right)$$
where γ is the surface tension of the liquid, ΔA is the change in surface area, r is nanopore radius, and h  is the consistent thickness. According to the Equation (1), it can be demonstrated that the nanopore with radius will reduce the surface free energy by decreasing, while nanopore with radius does so by increasing.

#### 2.1.1. Amorphous Nanopores

In 2003, the nanometer-scale modification was first realized in EB sculpting within TEM [33]. Beforehand, SiO_2_ nanopores with different initial sizes from closed pores to pores of about 200 nm were prepared by anisotropic KOH wet etching and thermal oxidation. Subsequently, taking advantages of TEM’s in situ imaging and direct feedback, the controllable shrinking process with a sub-nanometer accuracy has been entirely fulfilled by manipulating the electron-beam irradiation time under low electron intensity, which was ensuring the shrinking rate as lower as 0.3 nm/min [33]. Passing constant radiation, a shrinking nanopore was obtained in Dekker’s group from the initial diameter of about 6 nm to the final size of 2 nm only [33], as shown in Figure 2a–d. Meanwhile, the corresponding principle of minimum surface free energy was proposed, as shown above, which was then widely validated in high-energy-beam-induced modification in TEM.

Combined with the shrinking process, detailed studies on nanopore surface deformation has also been intimately concerned. Geometrical shapes in particularly their sharp curvatures were predominantly contributed to the variations. Due to the highest Laplace pressures induced by the strongest curves in the edges, all square nanopores were that the corners quickly round off [54]. Then the similar experimental phenomena were further confirmed in elliptical nanopores with a thickness gradient [55]. Apart from monitoring the nanopore surface, variations on cross-section have also received much more attention. Based on the most use of energy-filtering TEM (EFTEM) and electron tomography to 3D characterization, Qian et al. [56] observed that the nanopore shape changed from almost cylinder to hourglass after shrinking by the broad electron beam (beam intensity 9 × 10^5^ A m^−2^). The main reason originated from material migration occurred on both sides of the membrane at the same time, but on the substratum layer where the beam exited faster resulting in the narrowest part of the hole [56]. In particularly for the thinner film, the more pronounced transformation will be.

Recently, with the development of high-energy electron beams, Liu et al. [57] have found that EB can be alternatively used as a “paintbrush” to achieve controllable deformation on silicon nitride nanopores. Compared to the conventional approach, it adequately utilized the competitive relationship between the knock-on effect of the high-energy electron beam and surface tension-driven shrinkage. Without the limitation of in-site radiation and by flexibly moving the focus irradiation point of the electron beam from the initial positions, nanopores were deformed into any geometry with single nanometer accuracy. Typical geometric shapes such as squares, triangles, rectangles, T-shapes, as well as nano-slits about 2 or 3 nm width and 100 nm length have been widely acquired [57].

#### 2.1.2. Crystalline Nanopores

The field of EB-induced migration development has a rich and diverse choice of materials. Although the majority has relied on amorphous materials before, the incentive for entirely excellent performance on crystalline materials has also emerged. First, shrinking processes have almost exclusively carried out on Al_2_O_3_ nanopores from 4 nm to 1 nm only, in which the improved electrical properties (low noise, high signal-to-noise ratio (SNR)) was conducive to its application in single-molecule detection [34]. Secondly, further progress on crystalline magnesium and its alloys have made alternative strategies for nanopore modification. In contrast with the frequently-used round or nearly round holes, magnesium nanopores were surrounded by three groups of parallel crystal planes (four groups in magnesium alloys) [41,53], in which crystal plane shapes and sizes could be effectively controlled in different directions manipulated by the radiant direction and duration of electron beams owing to the anisotropy along crystal planes, as shown in Figure 2e–h. Furthermore, noticeable improvements including excellent electrical conductivity, stable mechanical strength and ductility [58], bright biocompatibility [59], as well as relatively low surface charge were increasingly practical to realize. However, an obvious setback in magnesium was local oxidation generated under the high electric field, which affects the uniformity of shrinking progress and the stability of nanopore morphologies apparently [41].

#### 2.1.3. Graphene Nanopores

As is well known, nanofilm thickness has a significant impact on SNR [60] and spatial resolution [61,62] in the concept of nanopore-based detection sensors. In 2004, the grand discovery of graphene provided a natural ultra-thin membrane inconceivable only 0.335 nm [63], which could realize single-base resolution in DNA sequencing theoretically. After six years of gradual improvement, suspended graphene nanomembranes have been successfully applied to drill graphene holes by EB for knocking the carbon atoms out of the hexagonal lattice of graphene, which was found out by three research teams simultaneously [64,65,66]. Since then, the controllable manufacture of graphene nanopores was gaining increasing attention. Until 2011, precise sculpting of graphene nanopores with atomic accuracy, which was accomplished in TEM and above 600 °C without destroying its crystal structure, has been demonstrated [67]. Compared to previous counterparts, shrinkage of graphene nanopores was expected to emerge extra at ultra-high temperature due to its self-healing mechanism to repair the crystal defects caused by electron beams via adsorbing carbon atoms [67]. Thereby alternative graphene nanopores from 7.4 nm down to 1.4 nm with finished edges have been fabricated as well as other carbon structures such as single-crystalline free-standing nanoribbons, nanotubes and single carbon chains [48,49], as shown in Figure 3a–g.

#### 2.1.4. Glass Nanocapillaries

In recent years, glass nanocapillaries have become a cost-effective source of nanopores for single-molecule detection [68]. Generally, they could be pulled with a laser-equipped pipette puller into tens of nanometers [69,70]. Despite this, much smaller diameters remained essential for high-resolution probing of single-molecule DNA and protein sensing. Hence, high-energy electron beams in SEM were further utilized to shrink nanocapillaries down to a few nanometers and the shrinking rate was fast reaching 0.25 nm/s, which could be altered by adjusting beam current or acceleration voltage [50], as shown in Figure 3h–j. A key advantage in the signal amplitude of DNA translocation detected by already-reduced glass nanocapillaries was 6 times higher in approximately rather than before. In comparison to silicon nitride, glass nanocapillaries also exhibited a better SNR [71]. In consequence, these considerable potentials have demonstrated that the emergence of nanocapillaries rather than conventional solid-state nanopores suspended on free-standing membranes has opened up a promising material in single-molecule detection.

### 2.2. FIB-Induced Material Migration

Atomic-scale arrangements were produced by high-energy ion beams when bombarding on a nanofilm by processes including material migration, sputtering erosion, ion implantation and so on [72]. The controllable fabrication in nanoscale made it a quite promising candidate in nanopore transformation. The main processes were combined with two competing mechanisms, material removal by ion sputtering and viscous material flow along sample surface, which were determined by ion fluxes. Therefore, high-flux dominating material removal was first used to drill large-sized nanopores and then directly switch the low-flux-induced material migration to shrink the nanopores controllably.

#### 2.2.1. Amorphous Nanopores

In 2001, ion beam sculpting was first visualized and named by Li et al. of Harvard University in order to shrink initial 100 nm silicon nitride nanopores down to 1.8 nm only in diameter [23], shown in Figure 4a–c. In this process, an equipped single-ion detector played a key role, which could monitor the real-time change of the transmitted ion current proportional to the veritable opening area and stop this process in time achieving nano-scale feedback control over pore sizes. In addition to dimensional variation, founders also presented their insight into the shrinking rate affected by sculpting time, sample temperature, duty cycle as well as ion flux, and mass [23]. According to a sufficient amount of experimental validation and simulation, the most use of heavy ions, appropriately increasing the sample temperature, lowering the ion flux and its duty cycle to a certain extent beneficial for the migration of surface substances and increasing the shrinking rate [73].

In addition, geometry nor the chemical surface composition determined the sensing performance of nanopore sensors without surface modification. As a result, researchers have gradually turned to nanopore’s three-dimensional structures during the shrinking process. Owing to far-ranging material migration, a volcano-like accumulating formation at nanopore opening was formed, which was detected by atomic force microscope [74], as shown in Figure 4d. The shape and size of these volcanoes, as well as the rate of expansion and shrinkage, were primarily affected by sample temperature and initial pore size [51]. Especially under lower temperatures, the formation of volcanoes accompanied by the shrinkage rate was slowed down or even stopped and slightly saturated at high temperatures. After that, the creation of volcanoes in other materials such as SiN_x_ [51], SiO_2_ [75], and a-Si [75] was also commonly observed. However, the accumulation of volcanoes inevitably brought about the increasing of nanopore thickness, which was an adverse effect on high-sensitive single-molecular detection.

After encountering setback in ion beam sculpting, the Harvard group transformed the original strategies into cold ion beam sculpting (CIBS) [76]. Similarly, only combined with extremely lower temperatures (less than 173 K), the surface material migration was effectively suppressed, which avoided the formation of volcanoes. Favorable results of ultra-thin silicon nitride nanopores with edge radiuses as small as 1 nm come true [76].

#### 2.2.2. Other Material

The novel and tunable structural features of nanoporous anodic aluminum oxide (AAO) has been intensively exploited for synthesizing a diverse range of nanostructured materials in the forms of nanodots, nanowires, and nanotubes, and also for developing functional nanodevices. Inspired by the former applications, the possibility of shrinkage was detected and contrasted between amorphous and crystalline AAO subjected to radiation of helium ion beam [52]. Two entirely different performances appeared: am-AAO nanopores finished to shrink within low flux before, while c-AAO presented remarkable resistance to irradiation most probably due to a barrier for atomic diffusion created by much higher ionic bonds. Subsequently, the scope of experimental materials was further expanded. For example, amorphous materials such as insulators (SiO_2_ and Si_3_N_4_), semiconductors (a-Si), and metallic glass (Pd_80_Si_20_) compared with crystal materials including Pt and Ag. Without a doubt, consequences were still the same as before. Therefore, more sufficient evidence suggested that dominance of FIB-induced migration tended to flow in amorphous materials only.

### 2.3. Other Methods

#### 2.3.1. Laser-Induced Material Migration

An intriguing method on shrinking nanopores in plastic material has been implemented by using the laser as an available heating source based on the surface-tension-driven viscous flow. Through the imaging on camera’s opposite side or detecting ion current in solution, they obtained real-time feedback of nanopore sizes, actually from several hundred microns to several hundred nanometers about 1000 times smaller [35]. Obviously, the study effectively extended the initial pore size range up to a few hundred microns rather than before and also broadened the alternative materials into low-cost plastic. Therefore, the most use of laser and other heating sources unveiled the potential of transformation on more high-viscosity low-melting materials. However, there was still room for improvement of size-control accuracy compared to other high-energy beams.

#### 2.3.2. Thermal-Heating Induced Material Migration

During the process of high-energy radiation previously reported, chemical composition around the nanopore surface was modified to a certain extent, which was disadvantageous for single-molecule detection owing to enhanced surface charge and electrical noise. Hence, a simple method of direct thermal heating emerged as times required. Without cumbersome sample-preparation steps and expensive equipment, only high-temperature treatment resulted in viscous flow and diameter decreased from 250 nm to 3 nm [36]. Furthermore, the shrinking process was strictly performed in the range of 1000 to 1250 °C. At low temperature (<1000 °C), the oxide layer would not be relaxed to an extent that it would start changing pore morphology. On the contrary, when the nanopores were processed at a higher temperature (>1250 °C), the oxide membranes either broke due to very high thermal stress or the shrinking process was too fast to control [36]. However, not difficult to find, problems of this method were using excessively thick SiO_2_ films (about 300 nm) and lack of visual feedback control on nanopore dimensions.

The effect of material migration has been widely observed in high-energy beams or direct-thermal heating. Based on the real-time visual feedback control, nanoscale dimensional modifications were entirely demonstrated at sub-nanometer precision. However, pronounced carbon deposition would inevitably contaminate the sample surface. In addition, limited processing space and exacting requirements in vacuum usually allowed only single-chip injection into the chamber. Therefore, improving manufacturing efficiency is still an essential challenge in material migration shrinking.

## 3. Shrinkage of Nanopore Surface Material Deposition or Growth

As discussed before, massive shrinking strategies are high-precision but time-consuming due to only one nanopore can be contracted at a time. After going nowhere, some researchers return to reduce pore sizes by depositing or growing new substances on the original nanopore surface without restrictions on pore number processed simultaneously. Remaining content followed by detailed discussion in different classifications. The comparison of their main properties is given in Table 2.

### 3.1. Material Deposition Shrinkage

Both nanopore surface properties and dimensions have been crucial in single-molecular probing and acquiring the desired SNR. Although there were great obstacles to realizing simultaneous control, deposition technologies still provide finishing processes in dimensional reduction and surface functionalization. Compared with other shrinking techniques, depositing procedures avoid changing or destroying the original morphology, but only deposit different materials, such as Al_2_O_3_, SiO_2_, Pt, Si_3_N_4_, etc. on nanopore surface selected by specific surface requirements. According to different shrinking principles, five deposition ways have been reported and summarized in the following sections.

#### 3.1.1. Atomic Layer Deposition

Atomic layer deposition (ALD) was performed as a standard method ready for depositing substances on the nanopore surface as a single atomic film layer by layer. During this period, the self-limiting growth where a newly atomic layer was directly related to a previous layer within the chemical reaction, ensured only a single sheet of atoms deposited per reaction [82]. Therefore, regardless of the initial size or shape of the nanopore, especially with a high aspect ratio, ALD can uniformly deposit new substances on all exposed surfaces while maintaining original morphology to shrink solid-state nanopores with single-atomic accuracy.

Based on these promising advantages, ALD was first carried out with Al_2_O_3_ onto the SiO_2_ surface. By adjusting the number of deposited layers, controllable shrinking of solid-state nanopores from 7.1 nm to 2 nm only was achieved entirely with extremely-slow deposition speed in 0.099 nm/cycle [38]. Furthermore, it rapidly expanded by means of widespread-use on oxides. Not only TiO_2_ [83], ZnO [84] could be deposited on the silicon nitride nanopores obtained by ion beam drilling, but Al_2_O_3_, ZnO [85], TiO_2_ [86], SiO_2_ [87] on porous anodized aluminum films with an ultimate aspect ratio of 1000 or a nano-porous polycarbonate film prepared by ion track etching [88,89,90,91,92].

Meanwhile, these experimental results, based on ALD, allowed researchers to broaden significantly the branches of application such as passivating non-ideal surfaces, eliminating the ion selectivity, reducing electrical noise, and also enhancing the DNA capture rate [93], which made it possible for desired single-molecule detection. For example, Kim et al. [94] used ALD to modify the nanopore surface with charge, which effectively increased the short ds-DNA translocation time. In addition, it remained a protective coating tool to improve the hardness and stability of nanopores [38].

#### 3.1.2. Vapor Deposition

Vapor deposition was a committed step for the well-established and versatile thiol chemistry used to functionalize metal surfaces [45]. This method came into immediate use by Wei et al. [45] depositing Ti/Au layer on Si_3_N_4_ nanopores where both shrinking and metalizing process was completely performed entirely at the same time. As they proposed, a linear relationship between shrinkage of pore size and thickness of the metal film was well guaranteed, by keeping the evaporation rate at a low level (0.7 Å/s), which was beneficial for controlling the shrinking process and predicting the final diameter down to 10 nm [45]. Mainly, these metalized nanopores had lower electrical noise in the liquid environment, and the speed of DNA passing through nanopore could also be efficiently slowed down by mutual attraction between DNA and metalized-pore-wall, which will significantly improve the collection of available signals during DNA sequencing.

Based on the former use, Wang et al. [39] soon switched to plasma-enhanced chemical vapor deposition (PECVD) depositing a Si_3_N_4_ layer on silicon nanopores with higher efficiency in shrinking rate, where all the nanopores from 50 nm to 400 nm were reduced to less than 10nm within 600 s [39]. Even so, a higher shrinking rate was equivalent to relatively lower controlling time and lower size-control accuracy (approx. several nanometers). Moreover, along with the accumulation of depositional time, uniformity of the Si_3_N_4_ layer gradually declined. In another word, PECVD was more suitable for the pre-shrinkage of solid-state nanopores [39].

#### 3.1.3. Ion-Beam Induced Deposition

Due to the lack of visual feedback, some researchers turned around to high-energy beams for nanopore shrinkage and localized functionalization. Compared with ion-beam sculpting, typical depositions were carried out with precursor gas first decomposed by secondary electrons reflected by ion beams bombarding sample surface, and then produced substance recombined and deposited on nanopore surface [95,96,97]. Using this strategy, the localized deposition of SiO_2_ has been directly achieved on silicon-nitride nanopore surface accompanied by a shrinking process from a starting diameter of 1 µm down to 25–30 nm in the end [44]. After that, various precursors have been attempted and integrated with nanopore shrinkage. In addition, conductive materials and insulating materials were also qualified via experimental verification with the final aperture below 10 nm [78], as shown in Figure 5a–c. Therefore, ion-beam induced deposition was an easily accessible approach to achieve nanopore shrinkage and localized functionalization synchronously with visual feedback control. Nevertheless, sophisticated chemical components around the nanopore surface made it difficult to conduct qualitative researches [78].

#### 3.1.4. Electron-Beam Induced Deposition

Branched from high-energy beams, electron beams were also been actively involved in decomposing and depositing processes named electron-beam induced deposition (EBID) [99]. Typically, SEM was first introduced to decompose hydrocarbon compounds and stored carbon on graphene nanopores, which was efficiently reduced to sub-10 nm [79]. However, the mechanism of electron beam shrinkage in SEM is still unclear, which hinders the fabrication of solid-state nanopores array as well as their applications. Recently, Yuan’s group revealed that the deposition of hydrocarbons dominates the shrinking process, accompanied by some decomposition of Si_3_N_4_. Based on this model, the smallest diameter of nanopore of 5.3 nm and the fastest shrinkage rate of 2.51 nm/s were obtained, which were assembled into solid state nanopores array [37]. Furthermore, SEM-induced shrinkage has also been continuously applied to silicon, silicon oxide as well as nano-slits [98], as shown in Figure 5d–h. In addition, on behalf of a higher degree of dimensional control, TEM was subsequently introduced to reach an ultra-low depositing rate only 0.6 nm/min combined with a sub-nanometer accuracy [100].

However, the carbon source of present works mainly originated from hydrocarbon pollutants in electron-beam chambers, which were neither stable nor reproducible. Therefore, due to lack of a continuous supplementary, Zeng et al. [101] recently focused on carbon tabs at the sample stage surface as a permanent carbon source, using SEM finish a linear size-reduction process. Meanwhile, it has also been further developed to silicon nitride, titanium nitride, titanium oxide as well as aluminum oxide nanopores. In brief, multiple sources such as internal contamination of the chamber (pumping system, specimen stage, stage lubricants, O-rings, etc.), conductive carbon glue, and even the sample itself could be regarded as carbon’s precursors. Therefore, under high-energy-beam radiation, a certain amount of specimen contamination on the sample surface was no means avoidable.

#### 3.1.5. Electrochemical Deposition

As mentioned before, most of the shrinking experiments reported in this review are performed in vacuum or atmosphere. To overcome this restriction, a pioneering experiment towards nanopore shrinkage performed directly in solution, by means of electrodepositing Pt, where platinum ions in the salt solution acquired electrons at the pore opening and became platinum atoms deposited at nanopore interface while monitoring its real-time ion conductance with feedback on the nanopore size, finally reduced below 20 nm [46], as shown in Figure 6a–c. Furthermore, this demonstration of electrochemical deposition may be extended to efficient parallel shrinkage of nanopore arrays and had an available access to depositing other metals than Pt. However, there was increasing concern about a series of unstable factors in the solution, such as local concentration, which may lead to unpredictable shrinking rates and uneven depositing surface [46].

### 3.2. Thermal Oxidation Shrinkage

In recent years, thermal oxidation technology has been entertaining in the present context as one of the critical steps for the controllable shrinking of inverted-pyramid silicon nanopore arrays, which were prepared beforehand by a combination of dry and wet etching [103]. More concretely, exposing to dry-oxygen oxidation at 900 °C, Si nanopore arrays on the pyramid surface was converted into SiO_2_, which were then soften and deformed with the feature size from 60–150 nm to sub 10 nm dependent on lower surface free energy [102]. Furthermore, surprisingly except the sharp edges of the silicon nanopores became round, the rest still maintained their typical inverted pyramid, as shown in Figure 6d–g.

The material deposition or growth is an efficient and parallel shrinking strategy, which usually contracts nanopore arrays or multiple chips simultaneously. Not only nanopore shrinkage but also surface modification could be completed via depositing alternative materials on the internal surface. However, the crucial issue exists in an inevitable increase in nanopore thickness, which will significantly affect the spatial and temporal resolution of single-molecule detection.

## 4. Recent Solutions for Small-Size Nanopore Fabrication

In this section, we will introduce some novel and recently used fabrication strategies of small-size nanopores (not limited to traditional shrinkage methods), so that you can find suitable manufacturing methods more comprehensively and quickly. First, small-size nanopores can be obtained by hybrid nanopores combining robust solid-state pores with natural small-size biological pores. Protein nanopores (5.4–6 nm in diameter) derived from thermostable viruses, electrokinetically inserted into larger-sized silicon nitride nanopores [104], as shown in Figure 7a. Second, in situ modification of nanopores by chemical methods is a handy method for fine-tuning the pore size. The TA/Fe^3+^ complex was initially adsorbed on the surface of the silicon nitride nanopore, and the complex crosslinking was increased by increasing the concentration of the crosslinking agent (Fe^3+^), which made the thicker film and the smaller size [105], as shown in Figure 7b. Third, ion channels can be obtained by controlling the interlayer spacing of 2D materials. Atomic-scale ionic transistors fabricated by multilayer reduced graphene oxide with a height of about 3 Angstroms exhibit ultrafast and highly selective ion transport [106], as shown in Figure 7c. Fourth, deposition techniques including in-situ electrochemical deposition and atomic layer deposition with sub-nanometer size-control precision have also been widely welcomed. Gold is electrochemically deposited on carbon nanoelectrodes, reducing the electrode gap to a quantum tunnelling space for molecular trapping and high-throughput analysis [107], as shown in Figure 7d. Atomic layer deposition of zinc oxide and aluminum oxide was utilized on the inner surface of nanopores to obtain thin-film transistors with diameter down to 10 nm for translocation detection of λ-DNA or bovine serum antibody proteins [108], as shown in Figure 7e. In addition, nanopores should not be limited to the preparation on nanofilms, but also can use photolithography to obtain the desired patterns on the lower platform, and obtain nanochannels by bonding the upper platform [109]. In this way, nanochannels with high aspect ratio can be easily prepared on the platform.

## 5. Conclusions and Outlook

This review summarizes useful insights from abundant shrinking strategies of solid-state nanopore combined with highlighted characteristics, which have been extended beyond size-morphology modification and dimensional control to surface functionalization. Given the current state of sub-nanometer precision with visual feedback control, high-energy beam induced drilling and sculpting, represented by EB and FIB, is still the most reliable manufacturing techniques, but restricted to valuable equipment and low processing efficiency. Therefore, peering only a few years into the future, a further cost-effective method should be increasingly emphasized and optimized for mass production of the nanopore-based platforms or devices with a uniform structure and outstanding performance given in DNA sequencing, single-molecular detection, energy conversion, sewage purification, and even organic degradation. Nanopore-based single-molecule sensors are set to remain active for years to come.

## Figures and Tables

**Figure 1 micromachines-13-00923-f001:**
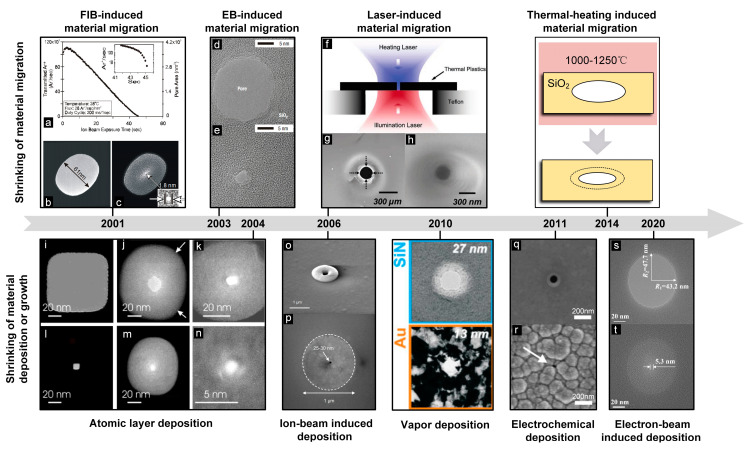
The timeline of typical shrinkage manufacturing technologies. Shrinking of original material migration: FIB-induced material migration [23], (**a**) Transmitted ion count rate and pore area, (**b**) 61nm pore, (**c**) after ion beam exposure; EB-induced material migration [33], (**d**) Before shrinking, (**e**) After shrinking; Laser-induced material migration [35], (**f**) A schematic, (**g**) Before shrinking, (**h**) After shrinking; Thermal-heating induced material migration [36]. Shrinking of new material deposition or growth: Atomic layer deposition [38], (**i**,**k**,**m**) Before shrinking, (**j**,**l**,**n**) After shrinking; Ion-beam induced deposition [44], (**o**) Before shrinking, (**p**) After shrinking; Vapor deposition [45]; Electrochemical deposition [46], (**q**) Before shrinking, (**r**) After shrinking; Electron-beam induced deposition [37], (**s**) Before shrinking, (**t**) After shrinking.

**Figure 2 micromachines-13-00923-f002:**
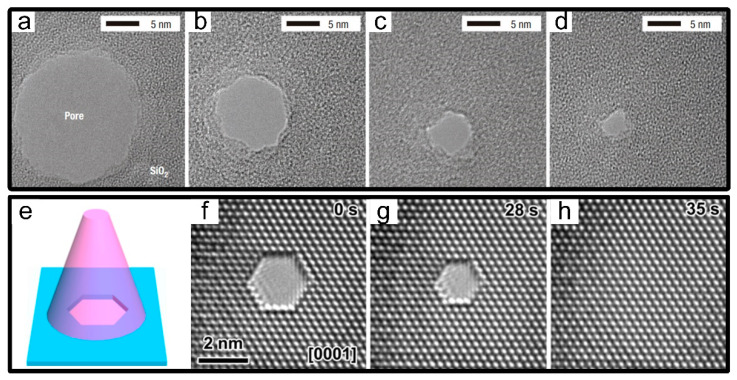
EB-induced material migration. (**a**–**d**) TEM images of 6 nm SiO_2_ nanopores were shrunk to 2 nm [33]. (**e**) Schematic illustration showing the self-healing of a nanopore under wide-field e-beam irradiation. (**f**–**h**) TEM images of continuous shrinkage of magnesium nanopores with an initial pore diameter of 3.3 nm under TEM radiation at 0, 28, and 35 s [53].

**Figure 3 micromachines-13-00923-f003:**
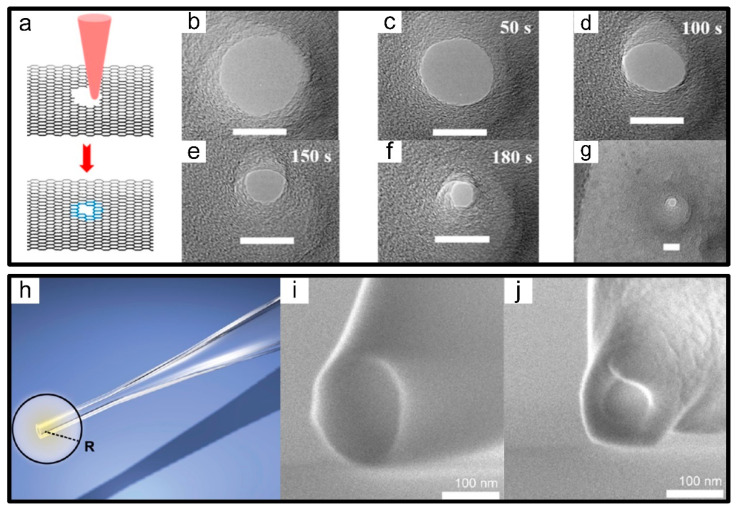
EB-induced material migration. (**a**) Scheme of the graphene nanopores shrinking under electron beam radiation. (**b**–**g**) Under the electron beam radiation of low current density, the continuous shrinking process of graphene nanopores. The scale bar is 15 nm [49]. (**h**) Scheme of the conical end of the nanocapillary. (**i**) SEM in-lens image of a quartz nanocapillary. (**j**) Shrunken nanocapillary after 14 min of irradiation under constant angle and beam parameters [50].

**Figure 4 micromachines-13-00923-f004:**
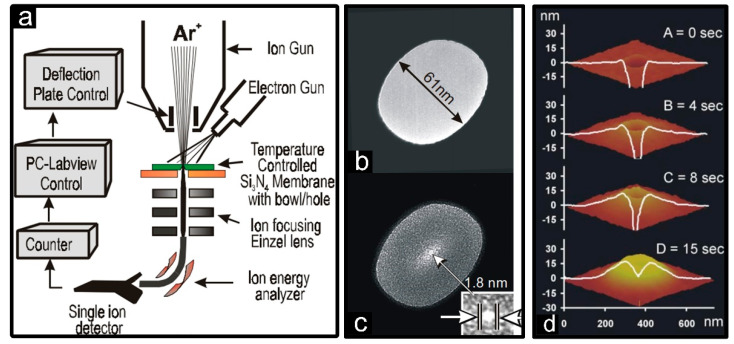
FIB-induced material migration. (**a**) Ion-beam sculpting system. (**b**,**c**) Argon ion beams sculpted Si_3_N_4_ nanopores from 61 nm to 1.8 nm [23]. (**d**) AFM surface scans after 0, 4, 8, and 15 s beam exposures [74].

**Figure 5 micromachines-13-00923-f005:**
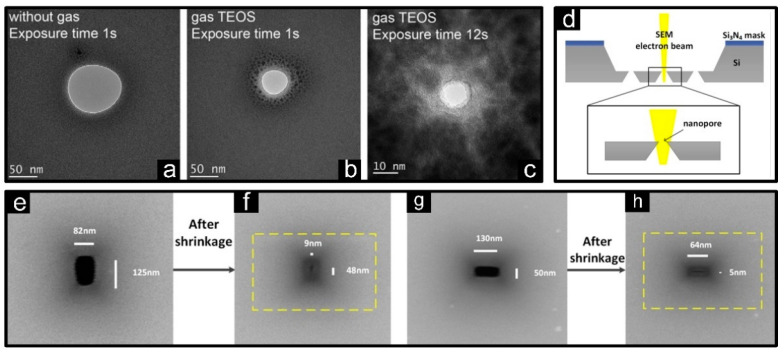
Shrinkage of material deposition or growth. (**a**–**c**) TEM images of nanopores fabricated by Ion-beam induced deposition with and without Si(OC_2_H_5_)_4_ (TEOS) gas for different exposure times [78]. (**d**) Schematic diagram of election-beam induced deposition. A nanopore before (**e**,**g**) and after (**f**,**h**) the EBID process [98].

**Figure 6 micromachines-13-00923-f006:**
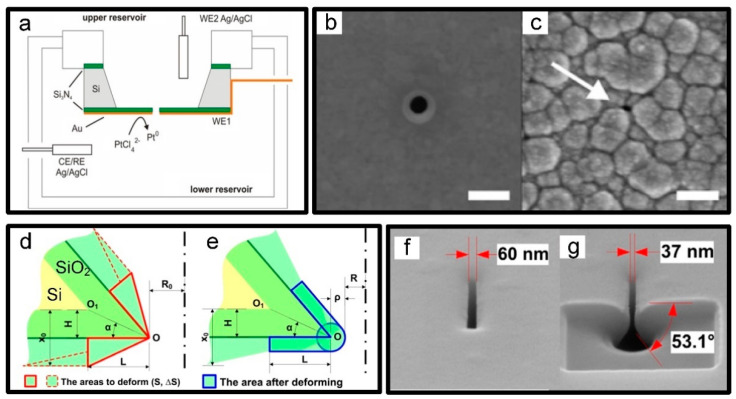
(**a**) Schematic diagram of electrochemical deposition. SEM images before (**b**) and after (**c**) electrodeposition (scale bar: 200 nm) [46]. (**d**,**e**) Schematic diagrams of the nanopore shrinkage induced by thermal oxidation: (**d**) growth of a SiO_2_ layer; (**e**) deformation of the SiO_2_ layer. (**f**,**g**) The inner structure of a nanopore before (**f**) and after (**g**) shrinkage [102].

**Figure 7 micromachines-13-00923-f007:**
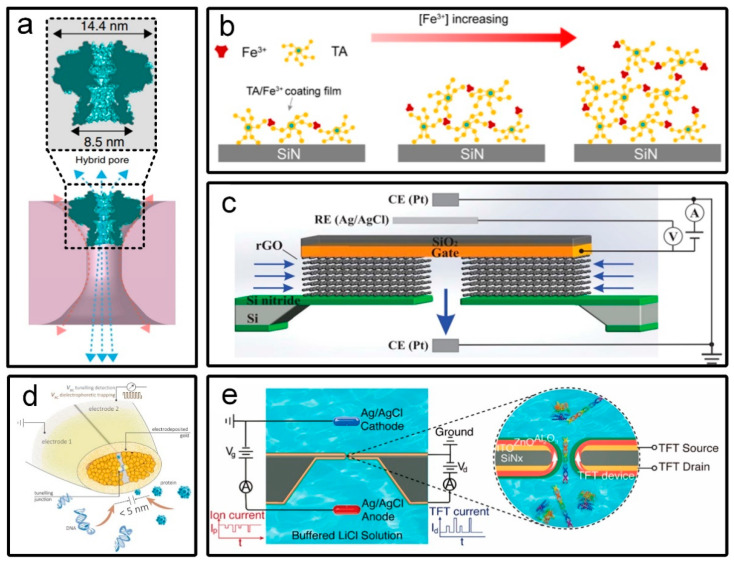
Recent solutions for small-size nanopore fabrication. (**a**) Cartoon image of the hybrid nanopore [104]. (**b**) Schematic diagram describing that the thickness of the coated TA/Fe^3+^ membrane is controlled by the concentration of the Fe^3+^ [105]. (**c**) Schematic of our atomic-scale graphene channel device with electric potential control using a three electrode configuration [106]. (**d**) Conceptual design of the quantum mechanical tunnelling probe [107]. (**e**) Schematic of a thin-film transistor−nanopore device and translocation experiment setup that concurrently measure both ionic and thin-film transistors signals [108].

**Table 1 micromachines-13-00923-t001:** Comparison of material migration shrinkage.

S. No	Shrinking Methods	Materials Tested	Minimum Diameter	Shrinking Precision	Reference
1	EB-induced material migration	SiO_2_	2 nm	1 nm	[33]
2		SiN_x_	2 nm	<0.5 nm	[47]
3		Al_2_O_3_	1 nm	Sub-nanometer	[34]
4		Graphene	1.4 nm	Nanometer	[48,49]
5		Glass	<10 nm	-	[50]
6	FIB-induced material migration	SiN_x_	1.8 nm	-	[23,51]
7		Am-AAO	<10 nm	Nanometer	[52]
8	Laser-induced material migration	Plastic	200 nm	-	[35]
9	Thermal-heating induced material migration	SiO_2_	~3 nm	Nanometer	[36]

**Table 2 micromachines-13-00923-t002:** Comparison of material deposition or growth.

S. No	Shrinking Methods	Materials Tested	Minimum Diameter	Shrinking Precision	Reference
1	Atomic layer deposition	Al_2_O_3_ on Si_3_N_4_	1 nm	~1 Å	[38]
2		TiO_2_ on Si_3_N_4_/TiN/Si_3_N_4_	1~2 nm	~1 Å	[77]
3	Vapor deposition	Ti/Au on SiN_x_	~10 nm	-	[45]
4		SiN_x_ on Si	<10 nm	Several nanometers	[39]
5	Ion-beam induced deposition	SiO_2_ on SiNx	25 nm	-	[44]
6		Pt, Si, C, etc. on SiN_x_	5 nm	-	[78]
7	Electron-beam induced deposition	C on graphene	<10 nm	-	[79]
8		SiO_2_ on SiN_x_	<10 nm	Sub-nanometer	[80]
9	Electrochemical deposition	Pt on SiN_x_	18 nm	-	[46]
10	Thermal oxidation	SiO_2_ on Si	8 nm	Nanometer	[81]

## Data Availability

Not applicable.

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
