# Peer review of "Controllable Shrinking Fabrication of Solid-State Nanopores"

_micromachines, 2022, doi:10.3390/mi13060923_

Round 1

Reviewer 1 Report

I have attached a PDF file.

Reviewer 2 Report

Nanopores have broad application prospects in the fields of DNA sequencing, protein sequencing, structure detection, etc. The fabrication of nanopores with small diameters is one of the most concerned issues. The authors introduce many methods for shrinking nanopores, and summarize the principles, advantages and disadvantages of shrinkage nanopore fabrication methods. However, minor revisions are still required before publication.

1. In the introduction, the author briefly introduces some methods and technological developments for shrinking nanopores. But a more specific background description is lacking. From the manuscript, the current fabrication methods and precision levels of solid-state nanopores are not introduced, nor are the differences between the different fabrication methods described in detail. Therefore, it is a question that needs to be answered whether it is necessary to study the shrinkage nanopore method.

2. As mentioned at the beginning of the manuscript, nanopore technology is an excellent sensor for molecular detection such as DNA and proteins. The manuscript also introduces many methods and results of shrinkage fabrication for smaller nanopore, but more detailed descriptions of experimental results are lacking, such as ionic current, noise, signal-to-noise ratio and other experimental results. The difference in performance of nanopore before and after shrinkage needs to be described. Otherwise, we have no way of knowing whether these shrinkage methods are meaningful.

Round 2

Reviewer 1 Report

The authors revised the manuscript point by point based on the reviewers' comments. This reviewer was satisfied with the revised manuscript. The tables and figures look much better. The authors have added a section on recent advances in nanopore shrinkage. The manuscript now sounds very positive for nanopore fabrication methods. The reviewer recommends this manuscript for publication.